# A Systemic and Local Comparison of Senescence in an Acute Anterior Cruciate Ligament Injury—A Pilot Case Series

**DOI:** 10.3390/life13071567

**Published:** 2023-07-15

**Authors:** Robert A. Waltz, Kaitlyn E. Whitney, Victoria R. Duke, Heidi Kloser, Charles Huard, Matthew T. Provencher, Marc J. Philippon, Chelsea Bahney, Jonathan A. Godin, Johnny Huard

**Affiliations:** 1Naval Health Clinic Annapolis, U.S. Naval Academy, Annapolis, MD 21402, USA; rawaltz@gmail.com; 2Linda and Mitch Hart Center for Regenerative and Personalized Medicine, Steadman Philippon Research Institute, Vail, CO 81657, USA; 3The Steadman Clinic, Vail, CO 81657, USA

**Keywords:** synovial fluid, peripheral blood mononuclear cells (PBMCs), senescence, senescence associated secretory phenotype (SASP), post-traumatic arthritis (PTOA), ACL

## Abstract

Background: Senescence, a characteristic of cellular aging and inflammation, has been linked to the acceleration of osteoarthritis. The purpose of this study is to prospectively identify, measure, and compare senescent profiles in synovial fluid and peripheral blood in patients with an acute knee injury within 48 h. Methods: Seven subjects, aged 18–60 years, with an acute ACL tear with effusion were prospectively enrolled. Synovial fluid and peripheral blood samples were collected and analyzed by flow cytometry, using senescent markers C12FDG and CD87. The senescent versus pro-regenerative phenotype was probed at a gene and protein level using qRT-PCR and multiplex immunoassays. Results: C_12_FDG and CD87 positive senescent cells were detected in the synovial fluid and peripheral blood of all patients. Pro-inflammatory *IL-1β* gene expression measured in synovial fluid was significantly higher (*p* = 0.0156) than systemic/blood expression. Senescent-associated factor MMP-3 and regenerative factor TIMP-2 were significantly higher in synovial fluid compared to blood serum. Senescent-associated factor MMP-9 and regenerative factor TGFβ-2 were significantly elevated in serum compared to synovial fluid. Correlation analysis revealed that C12FDG^++^/CD87^++^ senescent cells in synovial fluid positively correlated with age-related growth-regulated-oncogene (ρ = 1.00, *p* < 0.001), IFNγ (ρ = 1.00, *p* < 0.001), IL-8 (ρ = 0.90, *p* = 0.0374), and gene marker p16 (ρ = 0.83, *p* = 0.0416). Conclusions: There is an abundance of senescent cells locally and systemically after an acute ACL tear without a significant difference between those present in peripheral blood compared to synovial fluid. This preliminary data may have a role in identifying strategies to modify the acute environment within the synovial fluid, either at the time of acute ligament injury or reconstruction surgery.

## 1. Introduction

Post-traumatic osteoarthritis (PTOA) frequently develops after anterior cruciate ligament (ACL) and concomitant meniscal or other ligament pathology injury [1,2,3]. The incidence of PTOA after ACL injury is between 20.3 and 37% at 10 years, progressing to 28.6–62% after more than 20 years follow up [4,5,6,7,8]. Although few studies have shown that ACL reconstruction decreases the rate of PTOA, others have demonstrated no significant difference in the incidence of PTOA regardless of treatment, and there is no current evidence that surgical treatment can prevent its development and progression [6,9,10].

Synovial fluid profiles, which have shown cartilage extracellular matrix degradation and upregulation of inflammatory markers in the setting of intraarticular ligament injury similar to those found in chronic osteoarthritis (OA), may play a significant role the development of PTOA, fibrosis and graft incorporation [11,12,13,14]. Multiple PTOA animal models have evaluated molecular changes and mechanical overloading in both the cartilage and subchondral bone after ACL injury [15,16,17]. Additional studies have been performed specifically evaluating the synovium after ACL transection, and demonstrated upregulation of a pro-inflammatory response in the synovium, along with increased expression of both catabolic and anabolic genes [18,19,20].

Few human studies exist examining synovial fluid biomarkers at the time of both ACL reconstruction and meniscal repair/debridement surgery and, to our knowledge, there are no studies examining senescence in synovial fluid in the knee at the time of acute injury [11,21,22,23,24]. A more comprehensive understanding of this acute environment may play a role in identifying strategies for modifying the concentrations of detrimental biomarkers within the joint at the time of injury, to reduce the rate of articular cartilage degeneration and allow for improved healing.

Senescence, a characteristic of cellular aging and growth arrest in damaged or stressed tissue, and a critical component in controlled apoptosis for tumor suppression, has also been associated with OA [25,26,27,28]. Additionally, senescent cells release a complex inflammatory cascade known as the senescence-associated secretory phenotype (SASP), which is composed of pro-inflammatory extracellular proteases, cytokines, chemokines, and growth factors that, when persistent, can have a detrimental effect in homeostasis and repair, and are linked to both OA and PTOA [25,29,30,31]. Common SASP factors, and the upregulation of the senescence-associated cell cycle genes, *p16* and *p21*, have all been seen in chronic OA, and have been detected in cartilage, subchondral bone, and synovium in animal studies [29,32]. Additional animal studies have shown both an increase in senescent cell accumulation with cartilage injury and, conversely, clearance with the use of senolytics [26,29,33]. Targeted clearance of senescent cells through senolytics or inhibition of SASPs, such as inflammatory cytokines and specific matrix metalloproteinases present in cartilage injury through senomorphics, may be a way to reduce the incidence of PTOA [28]. To our knowledge, the presence of senescent cells, their associated SASP factors, and gene markers have not been evaluated in the synovium of acute intraarticular ligament injury in humans. Identification of senescent cells in human synovial fluid at the time of injury may provide increased understanding of the cartilage degradation process and facilitate a way to prevent this on a molecular level.

The purpose of this study is to prospectively identify, measure, and compare senescent profiles in synovial fluid and peripheral blood in patients that have sustained an acute ACL tear within 48 h. Secondarily, we aim to evaluate the senescent phenotype (senescent cell burden, SASP constituents, senescence associated gene expression) at the local versus systemic level. We hypothesized that senescent cells would be detectable in synovial fluid and in peripheral blood after an acute ACL tear, with no difference in the senescent phenotype. Additionally, the study aims to provide preliminary data as the first to detect and characterize senescent profiles in synovial fluid and peripheral blood after an acute knee injury.

## 2. Materials and Methods

### 2.1. Patient Enrollment and Demographics

This study was approved by our Institutional Review Board (protocol 2020-49). Patients between 18 and 60 years of age, who presented within 0–48 h of a primary, acute knee injury with significant intraarticular effusion and confirmed ACL tear on magnetic resonance imaging (MRI), were prospectively enrolled in the study between 8 February 2021 and 30 June 2021 (Figure 1). Demographic data, as well as body mass index (BMI), current medications, smoking status, medical comorbidities, and concomitant acute ipsilateral knee pathology on MRI were collected. Exclusion criteria included patients over the age of 60, prior ipsilateral ACL injury or surgical treatment, radiographic OA, previous or current fracture of the ipsilateral knee, intraarticular knee injection within 6 months of injury, history of immune or blood disorders, avascular necrosis, and communicable diseases such as HIV/AIDS or hepatitis C. Informed consent was obtained for study enrollment and prior to any research procedures.

### 2.2. Arthrocentesis Procedure

Synovial fluid samples were obtained at the time of injury (0–48 h) by performing an arthrocentesis procedure. The patient was positioned in the supine position on an exam table, and the lateral knee was marked at the level of the superior pole of the patella. A sterile prep was performed with chloraprep, and the skin was anesthetized with up to 5 mL of 1% lidocaine. After the superficial layers were anesthetized, additional chloraprep was applied to the knee, and an 18- or 22-gauge needle was inserted into the intraarticular space using a patellofemoral approach. Then, up to 30 mL of synovial fluid was aspirated into a sterile syringe with 3 mL of anticoagulant citrate dextrose, formula A (ACD-A) to prevent coagulation. The synovial fluid was then transferred to a 50 mL conical tube. A compression bandage was placed over the knee after the aspiration was performed.

### 2.3. Venipuncture Procedure

Venous peripheral blood was collected on each enrolled subject at the time of the arthrocentesis. Using a standard venipuncture or intravenous blood collection procedure, approximately 15 mL of peripheral blood was drawn in a syringe prefilled with 3 mL of ACD-A, and an additional 5 mL of peripheral blood was drawn into a red-top tube without any anticoagulant for serum isolation.

### 2.4. Sample Preparation

*Synovial Fluid:* The acellular and cellular layers of the synovial fluid were separated by centrifuging the sample at 1500× *g* and 20 °C for 10 min. If necessary, additional ACD-A was added prior to processing, to prevent clotting and cell clumping. A total of 300–1500 µL of supernatant was aliquoted into three sterile microcentrifuge tubes, and subsequently snap frozen at −80 °C for protein analysis using multiplex immunoassays. If red blood cells were visible in the cell pellet, the cellular layer was processed either using a red blood cell lysing buffer (Sigma Aldrich, St. Louis, MI, USA, Cat #: 11814389001) or a SepMate^TM^ tube (StemCell Technologies, Vancouver, BC, Canada, Cat #: 85460) with a Lymphoprep^TM^ (StemCell Technologies, Cat #: 7861) density gradient separation system according to the manufacturer’s protocol, until red blood cells were no longer visible. Cells were then counted and aliquoted into cryovials with TRIzol^®^ (Invitrogen, Waltham, MA, USA, Cat #: 15596018) for RNA isolation or CryoStor^®^ (StemCell Technologies, Cat #: 7930) with 1% anticoagulant for flow cytometry. Samples for RNA isolation were snap frozen at −80 °C, while cells for flow cytometry were frozen at the controlled rate of 1–2 °C/min. A minimum volume of 2 mL of synovial fluid was necessary for the sample preparation methods described, which did not allow for processing of small amounts of synovial fluid or diluted samples from contralateral normal knees.

*Peripheral Blood:* Peripheral blood mononuclear cells (PBMC) were isolated using the SepMate^TM^ tube with Lymphoprep^TM^ density gradient separation system, according to the manufacturer’s protocol. Cells were counted and aliquoted for RNA isolation and flow cytometry following the same protocols detailed for synovial fluid.

### 2.5. Flow Cytometry

We have recently developed flow cytometry-based protocols to detect senescent cells using the fluorescent compound C_12_FDG and the monoclonal antibody for human CD87 (also known as urokinase-type plasminogen activator receptor, uPAR). C_12_FDG is a compound that, when hydrolyzed by β-galactosidase, an enzyme upregulated during senescence, fluoresces at a wavelength of 514 nm [34,35], whereas CD87 (uPAR) was recently established as a cell surface protein that is broadly induced during senescence [34,35,36,37,38].

Synovial fluid and PBMC samples were rapidly thawed in a water bath and immediately diluted (4:1) in fluorescent activated cell sorting (FACS) buffer (Phosphate Buffered Saline (PBS), 5% FBS, and 5 mM EDTA Bio-Rad Cat #: 1610729). Cells were centrifuged, and counted to ensure a minimum of 2.0 × 10^5^ cells per 1 mL culture media (DMEM/F12 ThermoFisherScientific, Cat #: 11-320-033, with 10% FBS and 1% pen/strep ThermoFisherScientific, Waltham, MA, USA, Cat #: 15140148). Bafilomycin (Cell Signaling Technology, Danvers, MA, USA, Cat #: 54645, 100 nM) was added, and cells were incubated on a shaker (150 rpm) in 5% CO_2_ at 37 °C for 1 h. Half of each sample was then separated and stained with C_12_FDG (Abcam, Cambridge, UK, Cat #: ab273642, 6.5 µM), and the other half remained an unstained control. All samples were then diluted (2:1) with FACS buffer, centrifuged, washed, and resuspended in 100 µL PBS. Zombie aqua viability stain (BioLegend, Waltham, MA, USA, Cat #: 423102) was added per the manufacturer’s instruction, and samples were incubated at 4 °C for 15 min. Cells were then diluted with 1 mL FACS buffer, washed, and resuspended in 100 µL FACS buffer. 5 µL Human Seroblock (Bio-Rad, Hercules, CA, USA, Cat #: BUF070B) was added to all samples for 10 min at room temperature. Samples were transferred to ice, stained with CD87 (uPAR; ThermoFisherScientific, Cat #: 46-3879-42), and incubated at 4 °C for 30 min. Cells were then washed at 500× *g* for 5 min and resuspended in FACS buffer. A Cytek^®^ Northern Lights (16v, 14b) flow cytometer was used to quantify the number of live/dead, CD87 positive, and C_12_FDG positive cells present in each sample in triplicate. Gating strategies were developed in accordance with established methods [34,35,37,39] and analysis was performed using SpectroFlo^®^ software (version 2.1, Cytek Biosciences, Fremont, CA, USA).

### 2.6. Multiplex Immunoassay

Plasma, serum, and the acellular synovial fluid fractions were placed in a warm bead bath until just thawed, then centrifuged at 1000× *g* for 10 min to remove debris. Approximately 300 µL of each sample were transferred to microcentrifuge tubes for multiplex immunoassays on the Luminex^®^ 100/200^TM^ System (Luminex Corporation, Tokyo, Japan, Cat #: LX200-XPON-RUO). Analytes were measured according to the manufacturer’s protocols using magnetic bead kits (all from EMD Millipore) described in Table 1. Resulting analyte concentrations were then calculated using the Belysa^TM^ Immunoassay Curve Fitting Software System (EMD Millipore, Burlington, MA, USA, Cat #: 40-122).

### 2.7. mRNA Isolation and qRT-PCR

Total RNA was isolated and purified from synovial fluid cells and PBMCs using TRIzol™, and transcribed into complementary DNA using the reverse transcription cDNA synthesis kit qScript™ (QuantaBio, Beverly, MA, USA, Cat#: 101414-106), according to the manufacturer’s protocol. Expression of inflammatory and senescence markers was detected by quantitative real-time PCR (qRT-PCR) using PerfeCTa^®^ SYBR^®^ Green FastMix (QuantaBio, Beverly, MA, Cat#: 101414-280) on an Applied Biosystems StepOnePlus RT-PCR thermocycler (Applied Biosystems, San Francisco, CA, USA). Primers were designed using PRIMER-Blast (NCBI), and primer sequences are listed in Table 2.

### 2.8. Statistical Analyses

Power Analysis: Due to the exploratory nature of the study, a post hoc power analysis was conducted to evaluate the achieved power of the study based on the observed data, including the sample size, effect size, and significance level. To address our primary hypothesis to detect and measure senescence in synovial fluid and/or peripheral blood, we examined the Spearman correlation coefficient (ρ = 0.83) between *p16* gene expression, a widely recognized marker of senescence, and C_12_FDG^++^/CD87^++^ senescent synovial fluid cells. The post hoc power analysis revealed that the study achieved a power of 0.84 using a one-tailed test with a significance level of 0.05 and a sample size of 7 subjects. Based on these results, the study had a sufficient statistical power and sample size to assess the presence of senescent cells in synovial fluid, as confirmed by the expression of a known senescence-related gene. Power analysis calculations were performed using G*Power statistical software (version 3.1.9.6).

Flow Cytometry Analysis: Synovial fluid and PBMC results were paired by patient, and analyzed for senescence markers, defined as the percentage of live cells positive for C_12_FDG or CD87. Samples were measured in triplicate, and triplicates were averaged. Percentages of synovial fluid and PBMCs were measured positive for senescence using dim (+), bright (++), and total (dim + bright) senescence markers C_12_FDG and CD87. The combination of both senescence markers was used to measure the percentages of double positives for both C_12_FDG and CD87 in cellular synovial fluid and PBMC samples. Results were analyzed for normality, and normally distributed samples were tested with a ratio paired *t*-test (α = 0.05). Samples that did not fit a normal distribution pattern were analyzed with a Wilcoxon’s matched-pairs signed rank rest (α = 0.05). Statistical significance was defined as *p* < 0.05 using GraphPad Prism^®^ software (GraphPad Prism Inc., Boston, MA, USA, version 9).

qRT-PCR Analysis: Expression of genes relative to housekeeping gene GAPDH was determined using 2^−ΔCT^, Wilcoxon signed rank testing was performed on 2^−ΔCT^, and normalized gene expression is presented as log(2^−ΔCT^). All significance testing was performed with paired, Wilcoxon rank sum testing on non-transformed data using GraphPad Prism^®^ software (GraphPad Prism Inc., version 9).

Biomarker Analysis: For the multiplex analysis, a non-parametric mixed-effects ANOVA test was used due to the non-normally distributed data that contained values that were undetectable or below the threshold, to test the overall null hypothesis of equivalency among the synovial fluid, plasma, and serum supernatant samples [40]. Only analyte values that met 70% or greater detectability, within the detectable range for each factor measured in pg/mL, were included in the final analysis using GraphPad Prism^®^ software (GraphPad Prism Inc., version 9).

Correlation Analysis: A spearman rank-based correlation analysis was performed between senescent cell percentages, SASP, and senescent-associated and inflammatory gene expression analytes separately for both PBMCs and synovial fluid. Sample size allowed for detection of correlation coefficient magnitudes ≥ 0.8 with 80% statistical power. Correlation calculations and graphical figures were performed in RStudio statistical software (RStudio, version 4.1.1) [41].

## 3. Results

Twelve consecutive patients with acute ACL injuries confirmed on MRI and presented within 48 h of injury were evaluated; one was excluded due to age, one due to presence of osteoarthritis, and three refused knee aspiration. A total of seven subjects with confirmed ACL injury were enrolled in the study, with a mean age of 42.0 ± 15.3 years (range 21.4–60.7). Mean BMI (kg/m^2^) was 26.0 ± 2.8 (range 21.3–28.2). No patient used tobacco products. Patient demographic and injury details are summarized in Table 3. Patients were contacted at a minimum of 5 days after the venipuncture and knee aspiration procedures, and no adverse events were reported. The mean time from injury to synovial fluid and peripheral blood collection was 1.0 days, with a range <0.1 to 2 days.

Senescent cells were detected in the synovium and peripheral blood of all patients. These included both dim (+) and bright (++) expression for both C_12_FDG and CD87 (Figure 2). No statistically significant differences were found in the percentage of senescent synovial cells, compared to peripheral blood at a significance of α = 0.05. However, there were near threshold data observed in the comparison of CD87^+^ PBMCs and synovial fluid samples (Figure 2D, *p* = 0.0526). The majority of senescent cells detected were double positive for dim C_12_FDG and dim CD87 in both samples (white, Figure 2H–I).

Canonical pro-inflammatory genes (*IL-1β*, *IL-8*, *IL-6)* and the senescence-associated genes (*p53*, *p16*, *p21*) were measured in the cellular component of the synovial fluid and PBMCs. All data were log transformed to achieve normal distribution. *IL-1β* gene expression in synovial cells was significantly higher than PBMCs (Figure 3A, *p* = 0.0156). Relative expression of *IL-8* and *IL-6* was not significantly different in synovial fluid and peripheral blood. Similarly, relative expression of senescence markers *p53*, *p16*, and *p21* were not significantly different. (Figure 3B).

Multiplex immunoassays were also performed to compare the protein levels of senescent, inflammatory, and regenerative factors in the synovial fluid and blood (Figure 4). Only analytes that met 70% detectability were included for analysis, and presented results only include serum compared to synovial fluid, as plasma results were comparable to serum. While some significant differences were observed, there were no clear patterns associated with any of the specific categories. Consistent with gene expression, IL-8 concentrations in synovial fluid trended higher compared to serum and plasma, but there was no significant difference (Figure 4A). Interestingly, inflammatory biomarkers IFNγ and GRO trended higher in serum relative to synovial fluid, but were not statistically elevated (Figure 4B). Mixed results were noted in senescence-associated proteins, with synovial fluid showing significantly higher concentrations of MMP-3 (*p* = 0.0069) and trending higher in MCP-1 and IP-10. MMP-9 was the only senescence-associated analyte that was higher in serum than synovial fluid (Figure 4A, *p* = 0.0313). Similarly, consistent trends were not observed for the regenerative factors at the local versus systemic level; pro-regenerative MMP inhibitors TIMP-1 and TIMP-2, as well as the interleukin-1 receptor antagonist IL-1RA, were higher in synovial fluid than serum (Figure 4A, TIMP-1 *p* = 0.0625; TIMP-2 *p* = 0.04700; IL-1RA *p* = 0.0313); however, the angiogenic factor PDGF-AA (*p* = 0.0625) and growth factors TGFβ-1 (*p* = 0.0625) and TGFβ-2 (*p* = 0.0313) were higher in serum compared to synovial fluid (Figure 4B).

We next aimed to understand which factors most strongly correlated with in the senescent cell classification determined by flow cytometry, to determine which biomarkers most strongly drove the double bright C_12_FDG^++^/CD87^++^ phenotype and non-senescent C_12_FDG^−^/CD87^−^ cell phenotype. In the synovial fluid, the biomarkers that most strongly correlated with the C_12_FDG^++^/CD87^++^ phenotype included: GRO (ρ = 1.00, *p* < 0.001), IFNγ (ρ = 1.00, *p* < 0.001), and IL-8 (ρ = 0.90, *p* = 0.0374) (Figure 5A). The expression of the senescent pathway gene *p16* was also significantly positively correlated to C_12_FDG^++^/CD87^++^ senescent synovial fluid cells (Figure 5A, ρ = 0.83, *p* = 0.0416). Additionally, a significant positive correlation was observed between non-senescent C_12_FDG^−^/CD87^−^ synovial fluid cells and the regenerative factor IL-1RA (ρ = 0.90, *p* = 0.0149) in acellular synovial fluid (Figure 5B). Strong, but not statistically significant, positive correlations existed between non-senescent synovial fluid cells and acellular expression of the growth factors TGFβ-2 (ρ = 0.77, *p* = 0.0724) and TGFβ-1 (ρ = 0.71, *p* = 0.1108).

Similarly, in the peripheral blood, strong positive correlations were observed between a variety of senescent and inflammatory biomarkers in acellular serum and plasma with double bright senescent C_12_FDG^++^/CD87^++^ PBMCs, but no significant relationships were found (Figure 6A). Specifically, C_12_FDG^++^/CD87^+^ PBMCs positively correlated with the expression of inflammatory marker *IL-1β* (ρ = 0.60), plasma MCP-1 (ρ = 0.60), serum MCP-1 (ρ = 0.54), and serum MMP-1 (ρ = 0.43) (Figure 6A, solid bars). There were a variety of strong, but not statistically significant, negative correlations between C_12_FDG^++^/CD87^++^ PBMCs and SASP/inflammatory biomarkers: plasma TNF-α (ρ = −0.80), plasma MMP-3 (ρ = −0.70), plasma IFNγ (ρ = −0.70), serum MMP-12 (ρ = −0.54), serum GRO (ρ = −0.50), and gene expression of inflammatory marker *IL-8* (ρ = −0.60) (Figure 6A, striped bars). Additionally, strong correlations were measured between non-senescent C_12_FDG^−^/CD87^−^ PBMCs and the regenerative factor IL-1RA (ρ = 0.70) (Figure 6B, solid bars).

## 4. Discussion

The most important finding of this study is that there is a significant presence of senescent cells locally in the synovial fluid within hours after an acute intraarticular ligamentous knee injury involving the ACL. To our knowledge, this is the first study to detect senescent cells after an acute intraarticular ligamentous injury in a human. Within this case series, there were no statistical differences between senescent cell levels detected in peripheral blood and synovial fluid. This finding is impactful, as it may suggest that the systemic senescence readings acquired from blood can serve as an adequate proxy tissue to synovial fluid, which is far more difficult and painful to obtain through arthrocentesis.

By using two different senescent detection modalities (C_12_FDG and CD87), we were able to identify unique staining patterns and suggest that the combination of these two markers may help to categorize cells in various states of senescence. The composition of total senescence was driven (~54%) by a population of dim senescent cells (C_12_FDG^+^/CD87^+^), compared to the less frequent double positive C_12_FDG^++^/CD87^++^ cells (2% synovial fluid to 21% peripheral blood). While the science supporting senescent cell detection is rapidly expanding, the functional consequence of these different populations remains unclear. We propose that the C_12_FDG^++^/CD87^++^ cells may represent a subset of highly senescent cells, whereas the C_12_FDG^+^/CD87^+^ may correlate with a pre-senescent state. We speculate that the pre-senescent state may less deleterious or possibly associated with an acute injury response.

In support of our clinical findings, the presence of senescent cells has previously been reported in a transgenic mouse model following ACL transection [29]. Specifically, Jeon et al. identified the accumulation of p16^INK4a^ expressing senescent cells in the synovium and cartilage and further found that selective elimination of senescent cells slowed the progression of PTOA and increased cartilage development [29]. They found that selectively removing senescent cells from in vitro cultures of chondrocytes in arthritic patients undergoing knee arthroplasty decreased expression of senescent inflammatory markers and increased cartilage tissue extracellular matrix proteins [29]. Our study expands upon this work by detecting senescent cells in humans after acute ACL injury, as well as detecting senescent gene expression and SASP biomarkers.

In addition to the detection of senescent cells, we were also able to detect a wide number of biomarkers within the synovial fluid using multiplex assays, and we generally divided them into pro-inflammatory, SASP, and regenerative factors. Relative to peripheral blood, synovial fluid demonstrated significantly higher concentrations of MMP-3 and TIMP-2, with inflammatory marker IL-8 trending higher as well. Detection of these cells in animal models after induced ACL injury has been shown to have a correlation with chondrocyte damage, synovitis, and OA with similar findings also found after senescent cells were injected into the knees of mice [29]. Additionally, elevation of TIMP-1 and TIMP-2, known regulatory responses to elevated MMPs, may represent a complex pro-regenerative response from senescent cells after injury by trying to reduce SASP [14].

OA inflammatory mediators, such as IL-1, IL-6, and matrix MMP-3, are SASP factors seen in chronic OA and have also been identified in ACL deficient knees, indicating a potential link between senescent cells and acute injury [13,14,25]. Though we were able to find a significant elevation in both MMP-3 and TIMP-1 in synovial fluid, which are associated with cartilage proteolysis and repair, respectively, further investigation is needed to look at the trend of these levels several days out from injury, as well as their direct link to senescent cells and their associated SASP factors. Additionally, the negative correlation between regenerative factors TGFβ-1 and TGFβ-2 and dim and total synovial fluid senescent cells may represent a delayed reparative response in acute injury, or even senescent mediated down regulation of these regenerative factors within the first 48 h after injury. This also warrants further investigation in serial evaluation of synovial fluid in the same patient several days or even weeks after injury.

One animal study by Ayturk et al. used a mini-pig model to induce PTOA by transecting the ACL and looking at gene factors associated with proteolysis at different time points from injury [18]. They found the highest number of gene transcripts at day one, with significant elevations associated with both proteolysis and cartilage development between 5 and 14 days after transection, suggesting involvement of both catabolic and anabolic process within the synovium of an acute intraarticular knee injury [18]. Another study by Heard et al. looked at a similar animal model in sheep inducing PTOA through ACL transection and reconstruction, and found the highest levels of inflammatory molecules at 2 weeks, largely normalizing by 20 weeks [19]. In our study, we found *IL-1β* gene expression to be significantly higher in synovial fluid compared to peripheral blood, but no significant elevations in senescence associated gene markers of *p53*, *p16*, or *p21* in synovial fluid, though they did trend higher than peripheral blood. The detection of senescent synovial cells and associated SASP factors associated with both anabolic and catabolic cartilage processes in our study extends the findings of these prior animal studies to humans, and provides a basis for further investigation.

Though this study represents a small cohort of patients, it does present the first-of-its-kind human baseline data with comprehensive senescent phenotyping across two tissue compartments (synovial fluid and peripheral blood) following acute knee injury, and will form foundational data from which to build on in future investigation. Importantly, our data suggests no statistically significant difference in the quantification of senescent cells from the blood compartment compared to synovial fluid, which may increase the feasibility of conducting future longitudinal studies using blood only. Further investigation is warranted to evaluate temporal changes in synovial senescence and biomarkers at serial time points after ACL injury or after surgical treatment, and will assist in better understanding the complex synovial environment after injury and in development of senolytic treatment to potentially prevent PTOA.

We recognize this study is not without limitations, which includes low patient enrollment, particularly due to the logistical difficulty of capturing patients within 48 h of an acute intraarticular knee injury. Larger cohorts could increase the power and sensitivity. We were unable to establish a control group to evaluate synovial fluid senescent cells in normal knees, as the low aspiration volume attainable in a physiologically normal knee was too low for detection with current technology, and dilution of the samples was not comparable in pilot testing. Further exploration of differences between the bright versus dim senescent cell populations are needed to determine if they are functionally similar or different in the setting of acute knee injury and development of OA/PTOA. Additionally, we acknowledge senescence and SASP are complex, with both reparative and catabolic functions that will require further investigation to delineate the clinical implications of senescent cells.

## 5. Conclusions

There is an abundance of senescent cells locally and systemically after an acute ACL tear, without a significant difference between those present in peripheral blood compared to synovial fluid. This preliminary data may have a role in identifying strategies to modify the acute environment within the synovial fluid either at the time of acute ligament injury or reconstruction surgery.

## Figures and Tables

**Figure 1 life-13-01567-f001:**
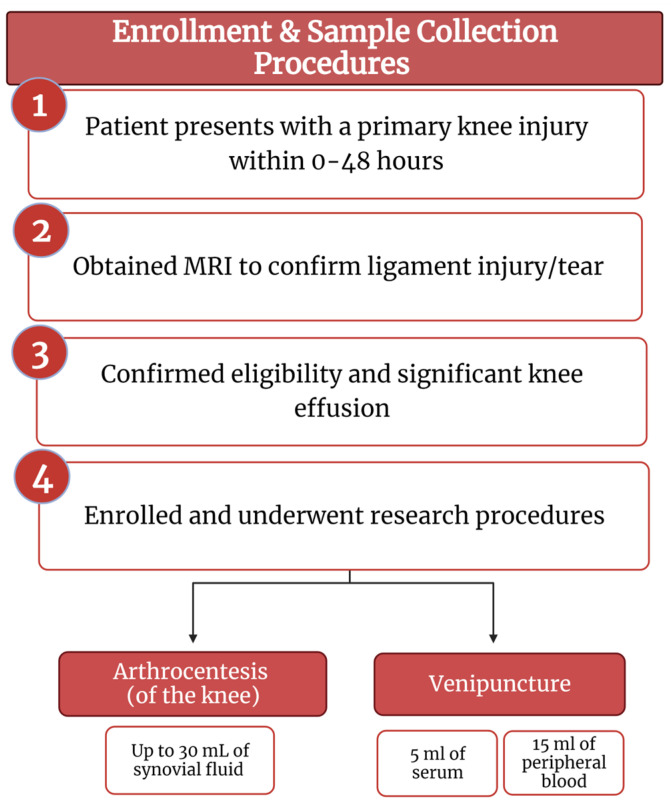
Enrollment and sample collection procedures flow diagram (created with BioRender.com).

**Figure 2 life-13-01567-f002:**
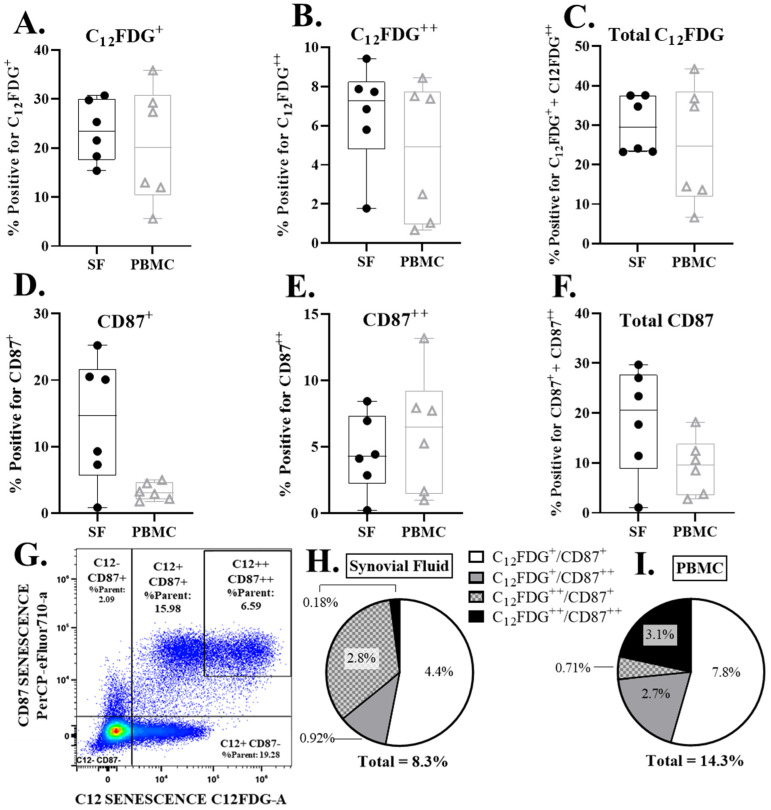
Senescent cell burden (CD87 and C_12_FDG positive cells) in PBMCs and synovial fluid (SF) via flow cytometry. (**A**–**C**) Percent dim (C_12_FDG^+^), bright (C_12_FDG^++^), and total (dim+bright) live, C_12_FDG positive cells in PBMCs and SF. (**D**–**F**) Percent dim (CD87^+^), bright (CD87^++^), and total (dim + bright) live, CD87 positive cells in PBMCs and SF. (**G**) Representative flow cytometry pseudo-color dot plot of overall senescence marker expression in live cells. Cells in the upper right quadrant are positive for both C_12_FDG^++^ and CD87^++^ senescence markers. Cells in the bottom left quadrant are negative for all senescence markers. (**H**,**I**) Pie chart of the average double positive senescence expression subpopulations in SF (**H**) and PBMCs (**I**).

**Figure 3 life-13-01567-f003:**
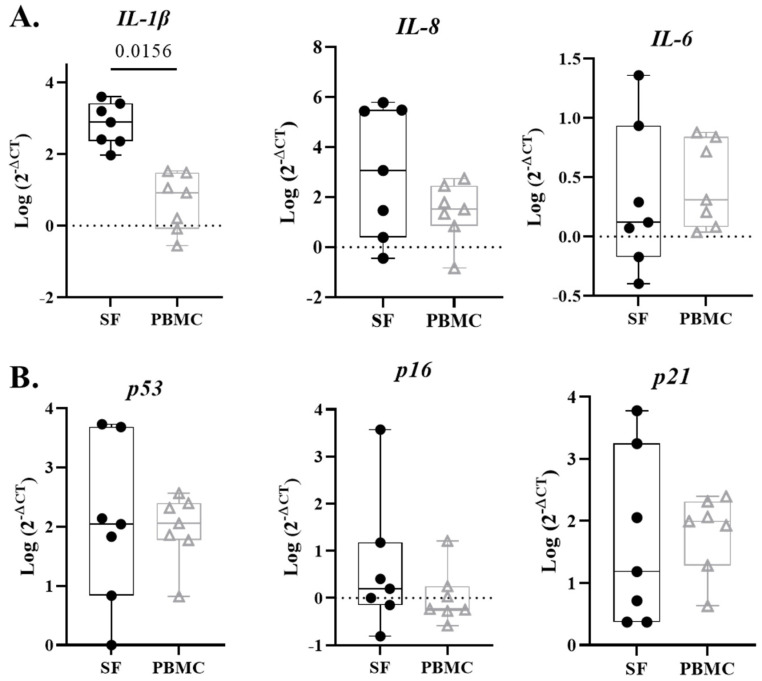
Senescent-associated and inflammatory gene expression in synovial fluid (SF) and PBMCs. (**A**) SF cells exhibit significantly more *IL-1β* compared to PBMCs at time of injury. Inflammation markers *IL-8* and *IL-6* do not significantly differ between synovial fluid and peripheral blood. (**B**) Senescence markers *p53*, *p16*, and *p21* did not demonstrated significant differences between SF and PBMCs.

**Figure 4 life-13-01567-f004:**
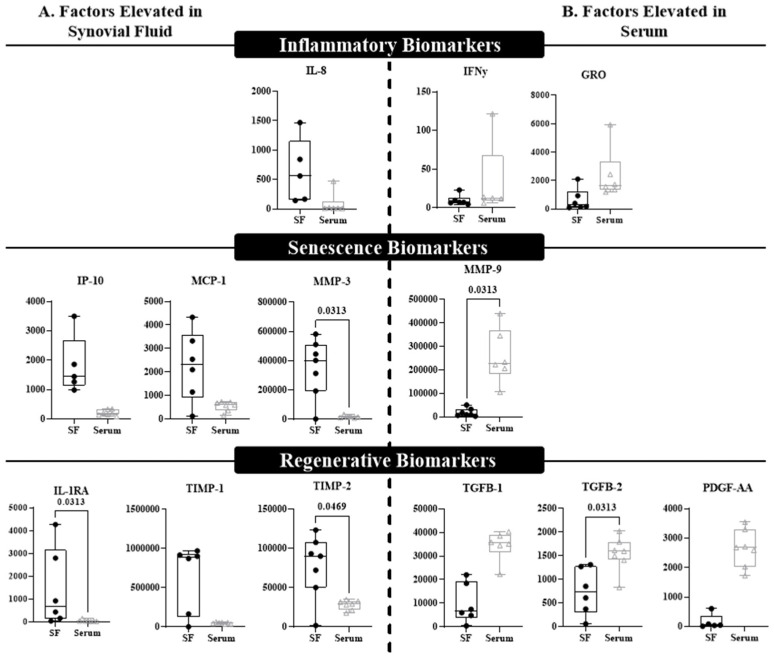
A multiplex assay distinguishes differential expression of biomarkers between synovial fluid and blood. Biomarkers that displayed difference at *p* ≤ 0.125 were included in this figure and divided by (**A**) factors that were elevated in the synovial fluid (SF) compared to blood (serum), or (**B**) factors that were elevated in the blood (serum) compared to SF. Significant differences (*p* < 0.05) are indicated within the graph. Each box plot represents the group median (middle horizontal line), interquartile range (IQR, top and bottom of boxes), range and outliers that exceed away from the nearest quartile. All graphs represent the concentration of the title biomarker in pg/mL.

**Figure 5 life-13-01567-f005:**
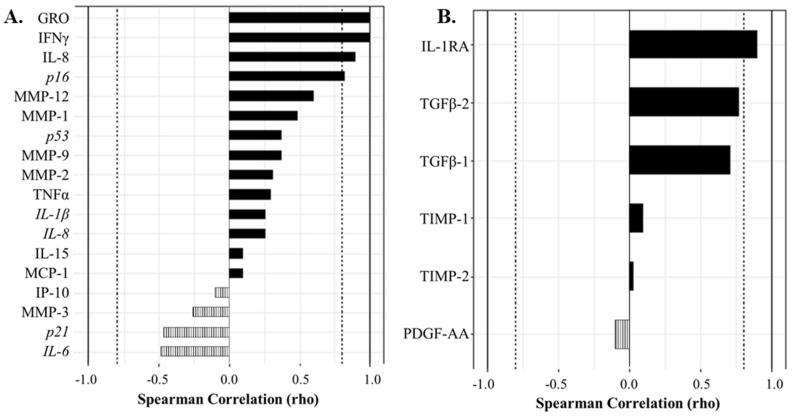
Correlation analyses show protein and gene factors drive the senescent and non-senescent phenotype in synovial fluid (SF). (**A**) Spearman correlation plot showing factors driving the strongly senescent C_12_FDG^++^/CD87^++^ cell phenotype. Genes are italicized, proteins are standard. (**B**) Spearman correlation plot showing protein factors driving the non-senescent C_12_FDG^−^/CD87^−^ phenotype. Dotted vertical lines represent traditional significance levels of ρ = ±0.80.

**Figure 6 life-13-01567-f006:**
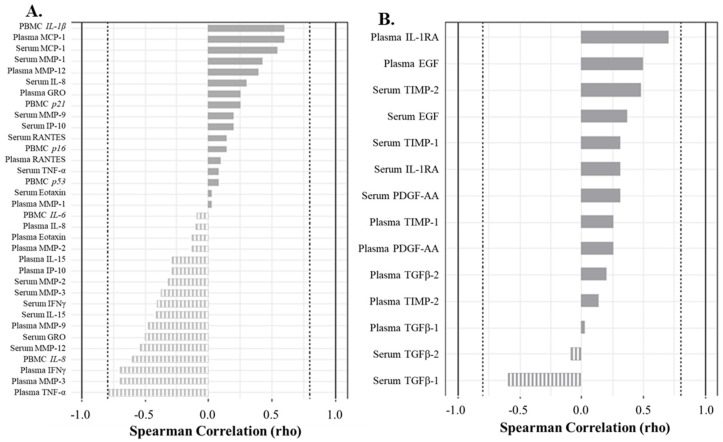
Correlation analyses show factors driving the senescent and non-senescent phenotype in blood (serum and plasma). (**A**) Spearman correlation plot showing factors driving the strongly senescent C_12_FDG^++^/CD87^++^ cell phenotype. Genes are italicized, proteins are standard. (**B**) Spearman correlation plot showing factors driving the non-senescent C_12_FDG^−^/CD87^−^ phenotype. Dotted vertical lines represent traditional significance level of ρ = ±0.80.

**Table 1 life-13-01567-t001:** Detailed outline of the magnetic bead kits used to perform the multiplex immunoassays on serum, plasma, and synovial fluid samples to detect SASP, inflammatory, and regenerative multiplex analytes. Abbreviated annalates are listed with their full description in the right column.

Kit Name (Cat #)	Abbreviated Name	Name
MMP panel 2 (#HMMP2MAG-55K-04)	MMP-1	Matrix metalloproteinase-1
	MMP-2	Matrix metalloproteinase-2
	MMP-9	Matrix metalloproteinase-9
MMP panel 1 (#HMMP1MAG-55K-03)	MMP-3	Matrix metalloproteinase-3
	MMP-12	Matrix metalloproteinase-12
Human cytokine/chemokine multiplex panel (#HCYTOMAG-60K-19)	Eotaxin	N/A
	EGF	Endothelial growth factor
	IFNγ	Interferon-gamma
	GRO	Growth regulated oncogene
	IL-15	Interleukin-15
	IL-1RA	Interleukin-1 receptor antagonist
	IL-8	Interleukin-8
	IP-10	Interferon-gamma-induced protein-10
	MCP-1	Monocyte chemoattractant protein-1
	RANTES	Regulated upon activation, normal T cell expressed and presumably secreted chemokine
	TNF-α	Tumor necrosis factor-alpha
	PDGF-AA	Platelet-derived growth factor-aa
TIMP panel 1 (#HTMP1MAG-54K-02)	TIMP-1	Tissue inhibitor matrix metalloproteinase-1
	TIMP-2	Tissue inhibitor matrix metalloproteinase-2
TGF-β universal kit (#TGFBMAG-64K-03)	TGF-β1	Transforming growth factor-beta 1
	TGF-β2	Transforming growth factor-beta 2

**Table 2 life-13-01567-t002:** Primer sequences used for qRT-PCR to detect inflammatory and senescent marker gene expression in synovial fluid and PBMCs. Interluken (IL), Glyceraldehyde-3-phosphate dehodrogenase (GAPDH).

Gene	Forward Primer	Reverse Primer
*IL-8*	5′-TTCTCCACAACCCTCTGCAC	5′-TCTGCAGCTCTGTGTGAAGG
*IL-6*	5′-TTCGGTCCAGTTGCCTTCTC	5′-GAGGTGAGTGGCTGTCTGTG
*IL-1*β	5′-GTACCTGTCCTGCGTGTTGA	5′-GGGAACTGGGCAGACTCAAA
*p16^INK4a^*	5′-CTTCCTGGACACGCTGGT	5′-GACCTTCCGCGGCATCTATG
*p21^Cip1^*	5’-CAAGCTCTACCTTCCCACGG	5′-ATCTGTCATGCTGGTCTGCC
*GAPDH*	5′-GCCTTCCGTGTCCCCACTGC	5′-CAATGCCAGCCCCAGCGTCA

**Table 3 life-13-01567-t003:** Patient demographics, injury details, comorbidities, and medications.

Patient	Age	Sex	Laterality	MRI Findings	Comorbidities	Medications
1	27.4	Male	Right	ACL tear, lateral meniscus tear	None	None
2	39.2	Female	Left	ACL tear, medial meniscus tear	Asthma, Vit D deficiency	Dulera, Ventolin, Vit D2
3	21.4	Male	Right	ACL tear	None	None
4	34.7	Male	Left	ACL tear	None	Ibuprofen
5	54.3	Female	Right	ACL tear	None	None
6	56.2	Female	Left	ACL tear, fibula styloid fracture	None	None
7	60.7	Female	Left	ACL tear, medial meniscus tear	Hypothyroid	Synthroid

## Data Availability

Data supporting reported results can be found at a data base at The Steadman Philippon Research Institute, Vail, CO, USA.

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
