# Peer review of "A Systemic and Local Comparison of Senescence in an Acute Anterior Cruciate Ligament Injury—A Pilot Case Series"

_life, 2023, doi:10.3390/life13071567_

Round 1

Reviewer 1 Report

Thank you for giving me the opportunity to review this well-designed and written study. It seems to me the study is not adequately powered to answer the research question. I suggest amending by mentioning the study design in the abstract and method sections. I have the following serious issues with this paper:

First, the authors did not indicate that how a sample size arrived at priority. The efficacy of the sample size is questionable. Whether the statistical analysis results with such a small sample size were convincing enough. I very much question the statistical validity of such a small sample size. Also, how authors determined data normality? It would be great if the authors clarify it.

Second, the experimental design of this manuscript is too simple, and the experimental data are very thin. It is suggested that after increasing the sample size, a richer statistical analysis should be conducted, and more valuable results should be obtained before considering publication.

Third, to justify publishing the work with the current sample size, the authors may consider presenting it as a preliminary report of ongoing research or short communication. This designation should be adequately captured in the title.

The authors have to elaborate on the connection between senescence and acute knee injury.

In general, adverse events will be documented for the last 30 days. In this study, it was reported for only 5 days. Is there any specific reason?

Suggestion: The authors should revisit their sample size determination and recruit more participants for the study to be sufficiently powered.

Minor editing of English language required.

Author Response

2 July 2023

Dear Editors and Reviewers,

Thank you for your offer of revision of our manuscript submitted as “Systemic and Local Comparison of Senescence in an Acute Knee Injury Setting.”  We would also like to sincerely thank the reviewers for the significant amount of time reviewing our manuscript and for providing very valuable feedback and suggestions. 

Our manuscript has been revised to the best of our ability to reflect the comments and suggestions of the reviewers with specific responses to the reviewer’s comments listed in the individual reviewer response letters.  We would first like to point out that we have modified the title to reflect reviewers comments and suggestions to more accurately reflect our study to “Systemic and Local Comparison of Senescence in an Acute Anterior Cruciate Ligament Injury – A Pilot Case Series.”

Despite encountering numerous challenges in study enrollment and data collection, our primary aim is to present preliminary evidence of senescent cell profiles in human synovial fluid and comparing local (synovial fluid) to systemic (peripheral blood) profiles. We firmly believe that the data has considerable value as it is intended to serve as a guiding framework to direct more robust prospective cohort studies. Further, this pilot case series provides initial insights into the detection methods of senescent cells in human synovial tissue after an acute knee injury. It is worth noting that this study is one of the first of its kind to showcase the presence of senescent cells in joint fluid subsequent to an ACL injury. Recognizing that an increased sample size would provide stronger statistical power to the study, we acknowledge the need to present our paper as a preliminary report. 

We thank you again for re-consideration of our novel and unique study.

Sincerely,

Johnny Huard, PhD                            Robert Waltz, MD

Reviewer 2 Report

Dear Authors,

Thank you for submitting your manuscript. I have carefully reviewed the article and found it an exciting contribution to the field. However, I believe a minor revision is necessary to improve the overall quality of the manuscript.

I encourage you to address these points in a revised version of your manuscript. This will significantly enhance the quality and impact of your work. Please do not hesitate to reach out if you have any questions or need further clarification on the revision suggestions.

Some suggested corrections have been listed below. Please answer point by point.

Introduction

-Lines 35-36: At the end of the sentence, "Post-traumatic osteoarthritis (PTOA) frequently develops after anterior cruciate lig-35 ament (ACL) and concomitant meniscal or other ligament pathology injury" I suggest including recent references that mention the importance of preventing PTOA after ACL reconstruction. I recommend you add these related references: doi: doi:10.1016/j.ocarto.2023.100366; doi: 10.1016/j.jor.2022.11.018; doi: 10.1186/s42836-023-00165-8

-At the end of Introduction, provide the exact hypothesis in detail when explaining the purpose of the study. The reader must also understand why your study is clinically relevant. What makes your analysis more appropriate than others already published? What is new about it? What makes your study necessary? What makes it worthy of publication?

Methods

-Respect abbreviations. There is some confusion. After mentioning an acronym for the first time, you can use only the abbreviation instead of writing the whole word. Correct the abbreviations in the text. Acronyms are not often given in the text; they should always be included the first time they appear to make the text more user-friendly. For example, line 80: Body mass index (BMI), etc.

-Please attach the "Institutional review board" or provide the study registration protocol number if available.

-Lines 128-155: This section is too wordy. It should be shortened without losing its meaning to make it more easily readable and interpretable to the reader.

Table 1, Table 2, Table 3: Add legend and abbreviation of abbreviated words (e.g., MMP...).

-Your measurement methods must be given in detail. What statistic program did you use? It would be good to add it to the text.

Results

-Was "Twelve patients" a consecutive series of patients? Please specify.

Discussion

-Well-written. Compliments!

References

-Add recommended recent references and, if possible, delete old and dated references.

Author Response

(The authors gave the same response as above.)

Reviewer 3 Report

The paper presents an interesting study on scenescence profiles in synovial fluid and peripheral blood in patients that have sustained an acute knee injury. The work is clearly discussed and presented.

Here are some minor suggestions to improve the quality of the paper:

1) Introduction: since the effects of senescence is an important aspect in the study, it is worth deepening it in the introduction, by comparing state-of-the-art studies on this topic.

2) Results: line 204-207 Did you perform a power analysis to verify whether your sample cohort is significant? Please mention this aspect in the materials and methods section.

3) Figure 5: a color code would be better to distinguish proteins and gene factors.

Author Response

(The authors gave the same response as above.)
